# Use of Wild Boar (*Sus scrofa*) as a Sustainable Alternative in Pork Production

**DOI:** 10.3390/ani13142258

**Published:** 2023-07-10

**Authors:** Antonia Lestingi

**Affiliations:** Department of Veterinary Medicine, University of Bari Aldo Moro, Valenzano, 70010 Bari, Italy; antonia.lestingi@uniba.it

**Keywords:** wild boar, wild boar expansion, wild boar management systems, carcass characteristics, meat quality, processed products

## Abstract

**Simple Summary:**

The impacts of agriculture in general and livestock farming in particular on environmental degradation have been causing increasing concern worldwide, especially in livestock production areas with a high animal density. However, advanced breeding and feeding strategies are needed to supply the growing global demand for animal protein. Wildlife might be a substitute for pork, beef, and poultry from intensive farming; this may be the case for wild boar, but the consumption rate is low. The main factors that would be responsible for the increase in the wild boar population and the scientific opinions of management are reported. This review aims to summarize the information currently available on the quantitative and qualitative characteristics of wild boar meat in order to stimulate its consumption, especially in processed products.

**Abstract:**

Pork production involves several sustainability issues. The recent increase in the natural wild boar population and the possibilities of its breeding to produce meat and for sport hunting have revived attention on this wild species. The most important factors that could account for its expansion and niche invasion are briefly summarized with the scientific opinion on management strategies. The information available to date on the quantitative, nutritional, and sensory characteristics of wild boar meat is reviewed to highlight its potential, if properly managed, as a sustainable option in meat production. This review reports on the opportunity of using wild boar meat in processed products and the need for research on processing qualities and acceptability for different final products. Above all, this review suggests that wild boar can be considered a sustainable alternative to meet the animal protein demand, as it can be established in marginal areas where it is already adapted to the environment, representing an interesting addition to traditional zootechnics.

## 1. Introduction

The interconnection between people, animals, plants, and their shared environment is a major concern today in light of the One Health approach [1]. This kind of approach is nothing new, but it has become more important in the last few years because of ever-growing increases in human populations, expansion into new geographic areas, changes in climate and land use, such as deforestation and intensive farming practices, and the increased movement of people, animals, and animal products due to international travel and trade, with increased risks of diseases spreading across borders and around the world [2]. As Lerner and Berg stated [3], One Health is a concept that involves the values of interdisciplinarity, public health, animal health, and ecosystem health. Furthermore, the One Health procedural model can be used to identify threats to health as well as positive models for the sustainable coexistence of humans and animals [1]. Cooperation between human, animal, and environmental health partners is essential to the success of public health interventions. Biodiversity and the conservation of geographical areas are among the health outcomes that can be achieved for people, animals, and plants using the One Health approach [1].

In light of these statements, sustainability has become a central goal for pig farming, as well as in agriculture in general [4]. Sustainability, a term frequently used in connection with biological systems, is defined as the ability of an ecosystem to maintain ecological processes, biodiversity, and productivity into the future [2]. Livestock farming is involved in global environmental degradation and global warming, through methane and nitrous oxide, especially in intensive systems [5,6]. In 2006, an FAO report stated [7] that the global animal industry was responsible for 18% of the total production of greenhouse gases (GHG; N_2_O, CO_2_, CH_4_, and H_2_O as expressed in CO_2_ equivalents (CO_2_e)), and the other parts are due to energy production plants, transport, industry, and environmental conditioning [2]. Methane (CH_4_) is a potent climate warmer that is often referred to as the second most important greenhouse gas (GHG) after carbon dioxide (CO_2_) [8]. In the first two decades after it is emitted, it is approximately 80 times more powerful than CO_2_ as a GHG [8]. Emissions and atmospheric concentrations of CH_4_ are continuing to rise [9].

Intensive meat, egg, and milk production systems are very often based on feeding grains and other ingredients sourced from far-off places, and this would make them economically and environmentally unviable if the price of feedstuffs rises above a critical level [10]. Moreover, more and more land has been converted to the intensive monocrop production of soybeans and corn (and other feed crops) globally, resulting in the pollution of waterways with pesticides and fertilizers, biodiversity reduction, the destruction of natural carbon sinks mainly due to direct and indirect land use change (dLUC and iLUC), and GHG emissions in all stages of intensive feed production and transport [2]. Intensive farming systems have been developed to achieve maximum production and profit, but they still contribute to ecosystem degradation and require careful examination regarding the excessive depletion of natural resources and the possibility of adopting more efficient farming systems [2].

In this review, several problems of intensive pig farming will be briefly highlighted in terms of sustainability and the need to mitigate the environmental impacts of this practice. After that, the review will turn its attention to the wild form of *Sus scrofa* that, in recent decades, has undergone a significant demographic increase, consequently invading new ecological niches, with negative effects on biodiversity, the health of the ecosystem, and human culture (i.e., the economy and social discontent). Scientific contributions regarding the main factors that would be involved in boar expansion and emergence will be explored in order to summarize the scientific opinion on management strategies. Further, the potential of farming wild boar as an alternative sustainable protein source will be highlighted, since the meat may come from slaughtered animals or animals living in the wild. In particular, differences in carcass characteristics and the physical, chemical, morphological, nutritional, and sensory properties of the meat between wild boars and domestic pigs will be reviewed in order to support the development of sustainable, welfare-positive production systems as a substitute for intensive pork production.

## 2. Intensive Pig Farming

Pig farming provides an important contribution to world food production, especially in Asia, accounting for around 30% of meat consumption worldwide [11]. The European Union (EU) is the world’s second largest producer of pork and the major exporter of pork and pork products [12,13]. Pig farming occurs with large differences in breeding systems as well as in the size of farms, with outdoor pig farming accounting for more than 16% of the total number of European pig farms and around 0.7% of the total number of pigs [12,13]. The negative impacts of pig farming on the environment are essential concerns, as there are several issues in terms of sustainability [4]. It has been reported that pig farming is globally responsible for 668 megatons CO_2_-e annually, representing nine percent of emissions from the livestock sector [10]. It has also been reported that overall greenhouse gas emissions from pork production and consumption is 12.1 kg CO_2_ per 1 kg and includes processing, domestic transport, retail refrigeration, home cooking, and waste disposal [5]. In comparison, the greenhouse gas emissions produced by 1 kg of lamb and beef are 39.2 CO_2_ and 27.0 CO_2_, respectively [5]. The carbon footprint and land use values for pork are 5 kg CO_2_-e kg^−1^ meat and 40–75 m^2^ kg^−1^ protein per year, respectively [14]. It has been reported that a globally sustainable diet entails a reduction in pork production by 85% [4]. Feed production and manure management (storage and spreading) in particular, as well as land use change from soy production, produce a critical load on the environment [2].

Due to the scale economies, larger units have been developed that can produce pork more cost-effectively [15]. In order to improve the economic efficiency of pig farms, housing systems have been developed (e.g., farrowing crates, fully slatted floors, and reduction in space allowance), which are not respectful of the normal behavior of pigs [12,16]. Tail biting [17] and stereotypic behavior [18] are among the problems that have arisen.

Developing countries are rapidly moving pork production to the industrialized model [2]. According to Gerber et al.’s (2013) [10] modern feed strategies that include beneficial feed additives like enzymes, amino acids, and gut modulation products, manure management practices and energy use efficiency reduce emissions related to farming. Kaufmann (2015) [2] has reviewed studies on the negative environmental impacts of pig farming, with special emphasis on a Chinese one, providing an integrated low-emission farm (LEF) concept, that is an innovative combination of nutrient, emission, and waste management, not only significantly reducing the environmental impact, but also improving the economic aspect of farming by producing renewable energy (heat, electricity, and biomethane) with animal manure as major feedstock in an anaerobic digester.

## 3. Habits, Life, Expansion, and Emergence of Wild Boar

The contexts of wild ungulate overabundance in Europe are protected, hunting, forestry, arable farming, livestock farming, and peri-urban areas [19]. Wild boar (*Sus scrofa*) also known as the wild swine, common wild pig, Eurasian wild pig, or simply wild pig, can be considered the progenitor of the domestic pig (*Sus scrofa domesticus*). This species is categorized by the International Union for Conservation of Nature and Natural Resources (IUCN) as one of least concern; it is one of the most widely distributed mammals throughout the Palearctic region, North Africa, and Southeast Asia, and it is currently only absent from Antarctica [20,21,22,23]. It has been successfully introduced in many countries, becoming naturalized (named feral hog) in the New World and Australia [20]. To date, human activities such as hunting, introduction into new areas, breeding in the wild, and businesses involving swine (i.e., the marketing of animals for meat) have affected the distribution of wild boar all over the world [24,25,26,27]. Such activities have influenced the geographical distribution of the species by moving animals and promoting hybridization events, affecting the natural potential of its dispersion, and allowing the species to occupy areas that probably could not have been reached by Suidae [20]. It is found on many continents with different ethnic groups [20]. High adaptability allows wild boar to inhabit a wide variety of habitats (e.g., Mediterranean scrubland, semi-desert, tropical rain forests, grasslands, and anthropogenic habitats) [28,29,30]. Wild boars’ diet, depending on the season, can consist of beechnuts, acorns, chestnuts, bulbs, rhizomes, and mushrooms; but, being an omnivore, foods of animal origin are not excluded and are based on local availability [20,23,31]. Acorns (*Quercus* spp.) represent a large part of the diet of wild boars (*Sus scrofa scrofa*) in their native European habitat [32].

Furthermore, through rooting, wild boars look for larvae, chrysalises, earthworms, and insects and if necessary, they will also eat carrion [20,31]. Overall, however, due to its size, it is an animal with low nutritional needs; it does not have any difficulty in feeding on agricultural products such as cereals, grapes, potatoes, etc. [31]. Its invasiveness depends on the obstacles it can encounter such as the presence of road networks, car collision events, and natural predators [33,34]. The natural predators, which depend on its geographical area, are wolves (*Canis lupus*), bears (*Ursus* sp.), leopards (*Panthera pardus*), striped hyenas (*Hyaena hyaena*), Eurasian lynx (*Lynx lynx*), bobcats (*Lynx rufus*), mountain lions (*Felis concolor*), and eagles (especially for piglets) [35,36]. However, even if considered harmful, in areas with the right agro-forestry density, wild boar is to be considered useful to the wood since, with its rooting and feeding on larvae and chrysalises, it moves the soil and buries the seeds of different forest species, thus favoring not only their germination and the birth of new plants but also the aeration of their roots [31]. At the same time, it removes a series of parasites ranging from insects to mice, thus contributing to the maintenance of the biological balance of the ecological niche to which it belongs [31]. It has been reported that wild boar may have added value as an ecosystem engineer for the preservation of threatened butterfly species [37,38,39]. Ecosystem engineers, such as large ungulates, are frequently used in conservation efforts in order to maintain biodiversity of open habitats [37]. Nevertheless, wild boar have been reported to be one of the top 100 species of the world’s most invasive and parasitic animals, even in its native range [40,41], raising many management issues for agriculture, economy, human and animal health, biodiversity, and human–wildlife relationships [23,42,43,44,45].

In recent decades, wild boar has undergone a significant demographic increase causing economic and biodiversity damage in various countries [13,19,20,24,46].

Several management plans have been employed by various countries to mitigate the problem of the growing expansion of the different wild forms of *Sus scrofa* (wild boar, hybrid, and feral pigs) into new geographical areas and ecological niches [47,48].

The relative importance of several factors involved in boar expansion has been highlighted in [20] in order to provide new insights for the research and the development of collaborative management policies.

The most decisive factors in determining the widespread intensifying of wild boar densities include climate change, human-induced habitat modifications, abandonment of rural lands, increases in forests and hunting areas, predator regulation of the prey, hybridization with domestic forms, and transfaunation [20,24]. Wild boar emergence depends on a combination of causal factors which could act simultaneously and influence each other [20].

## 4. Socio-Economic Activities Involving Wild Boar

Ungulates are involved in different socio-economic activities [19]. Protected areas are mainly dedicated to conservation, so in this context, the hunting of ungulates is limited to managed culling or, in a minority of cases, prohibited (e.g., national park) [19].

Furthermore, wild boar is part of wildlife used for producing meat and in sport hunting all over the world [24]. Hunting areas are characterized by land where the main human activity is hunting, which is carried out for commercial interests. Hunting management is an agricultural land use of great economic importance in Europe [49]. European boar hunting is carried out with the help of hounds and using rifles [50]. The main objective in hunting areas is to produce high-quality game animals and large trophies [51,52,53]. Wild ungulate husbandry for hunting restocking must provide animals with high trophy merit and good wild traits [54]. The main product are adult males with high trophy merit, the value of which largely surpasses that of the meat, which is considered as a side product [54]. In general, hunting areas have two different management regimes, managed estates (fenced or open) and unmanaged estates [19,55], with different possible situations present in both regimes.

In fenced hunting estates, the main management actions are perimeter fences, supplementary feeding [56], and, less frequently, the addition of single ungulates to alleviate inbreeding [57]. In Italy, restocking game is usually reared in fenced park-type settings with minimal input [31,53]. Wild boar is generally kept in mono-specific stocks, while the other species can be reared in mixed herds [54]. Border fences guarantee the captivity of animals, as required by the law for the conservation of the genetic integrity of indigenous populations [31,54]. Animal restocking generally includes game for the restocking of shooting preserves (wildlife hunting and private agri-tourism hunting lands). Shooting preserves often comprise large enclosures, used both for raising wild ungulates and hunting them during the harvest season. This extensive husbandry system is suitable for less favored areas, where wild ungulates can exploit scarce and poorly available resources more efficiently than domestic animals [31,54]. The livestock load is kept extremely low, between 1 and 100 hectares per 100 kg of live weight, and annual or seasonal hunting must be scaled accordingly [31].

In open-managed game estates, the most common hunting area regime in Europe [58], the most common management measure is winter feeding, which is associated with maintaining high hunting densities and improving the quality of the trophies [59]. Reduced antler size [60] may constrain economic interests in hunting areas.

Recent natural population increases and the potential for breeding have renewed interest in wild boar as a meat producer [24]. Rusticity and adaptability, prolificacy (3–8 piglets per birth), the relative simplicity of rearing, and meat quality are the drivers of interest in wild boar meat farming systems (Figure 1, personal photo), as a purebred stock or crossed with pigs [31].

Wild boar breeding has spread to Japan and several countries in Europe and North and South America [24,61,62]. Farming would ensure a constant supply of this type of meat throughout the year [5,24]. Deer and wild boar are adaptable and respond well to intensive management systems [31,32,54].

Farming of wild ungulates to meet the growing demand for animal protein in the human population of developing countries, such as South Africa, is a global reality in confined, semi-confined (wildlife farming) [63], and extensive systems (game ranching) [64,65].

In Italy, because of the climate and pastures and the pressure for land use, intensive systems are not common, with local exceptions regarding farmed wild boar [54]. Wild boar breeding is generally regulated by the same laws that apply to domestic pigs [31]. The ideal location for semi-extensive farms for meat production are wooded areas with broad-leaved trees (chestnut, downy oak, and holm oak), tall trees, and copses [31]. For farmed wild boar meat production, heavier animals are preferred, with higher growth rates and thus characterized by greater feeding requirements [31].

Compared to extensive systems, meat production farms require specific inputs (fencing and handling systems, supplementary feeding) and management programs, aimed at meeting farm animal welfare requirements, ensuring operator safety, and maximizing the rates of reproduction and weight gain, together with the production of quality products, in order to compensate for the relatively high fixed costs and achieve maximum profitability [54]. Part of the livestock area (from 3 to 10%) must be made up of arable land (potatoes, corn, etc.) and pastures. The presence of water must be ensured such as ponds, streams, or puddles. Animal density is established on the basis of the trophic resources and generally, a load of three animals per 10 hectares is implemented [31]. The required area for a farm of this type is very extensive and can vary between 100 and 1000 hectares. The fence represents the greatest investment to be made [66]. Handling facilities should allow agricultural activities such as: sorting into groups, loading onto transport vehicles, tagging, condition scoring, weighing, drenching, and vaccination [31]. Moreover, wild ungulates may be slaughtered in the crush with a captive bolt according to veterinary policy regulations, after vet ante-mortem inspection [54]. As is the case for those shot in the field, these animals are immediately bled on the farm and then transferred to a local licensed abattoir, where evisceration, dressing, and vet post-mortem examination are performed [54].

## 5. Wild Boar Carcass Characteristics and Meat Quality

Globally, wild boar is a primary food resource, mostly for remote rural communities outside of large urban centers located in Latin America, Asia, and Africa [24,63]. In Chile, the production of meat from European wild boar (*Sus scrofa* L.) is expanding together with the export of this “exotic” meat [67]. Furthermore, in the Finno-Scandinavian peninsula, and central and Mediterranean Europe, wild boar (*Sus scrofa*) and other game species meat (reindeer, red deer, roe deer, fallow deer, moose, and chamois) can be found in local restaurants and fairs, indicating an already consolidated gastronomic interest in this type of meat [63,68,69,70].

The price per kg of wild boar meat is rapidly increasing and the online sale of frozen European wild boar meat is currently EUR 26.56/kg (online search 22 June 2023). However, prices may vary in different countries and in different regions of each country (personal survey in Italy).

### 5.1. Carcass Characteristics

Studies concerning slaughter yields and meat quality of wild ungulates are numerically more limited than those carried out on domestic species. Furthermore, there is considerable variability among results obtained in different studies concerning wild boar meat production potential and its quality, due both to environmental, nutritional, and genetic factors as well as the adoption of different slaughtering and cutting systems for the carcasses [24].

Low growth rates are the main limiting factor for wild boar meat production [71]. Different slaughter weights between wild boars and domestic pigs (including crosses) can be explained by karyotypes, i.e., the different number of chromosomes (2n = 36, 37, or 38) due to chromosomal translocations (chromosomes 13 and 17 or 15 and 17) [72]. European purebred wild boars have 2n = 36 chromosomes, while wild boars from Eastern Europe and Asia, as well as domestic pigs, have 2n = 38 [73]. The different karyotypes lead to high values of live weight, food intake, and fat deposition and low meat conductivity values [74], quantitative carcass characteristics, and fat depth in *Sus scrofa* [75]. The proportion of fat in the carcass of wild boar with the 2n = 36 karyotype was lower (17.7%) than in phenotypically similar crossbreeds (2n = 37 and 2n = 38; 22.7%) [72]. Moreover, in the study by Skewes et al. (2014) [76], wild boars with 2n = 37 and 2n = 38 karyotypes had higher final live weights (270 days; 80.1 and 80.4 kg, respectively) than 2n = 36 (47.2 kg). The average daily weight gains were also higher in 2n = 37 and 2n = 38 crossbreeds (400.2 and 416.8 g day^−1^, respectively) than in the 2n = 36 group (246.7 g day^−1^), which showed values very similar to those reported for the European wild boars (range between 160 and 233 g day^−1^) [67,76].

It has been reported that increased cortisol values could be an additional factor contributing to a slower growth rate in wild boars than in domestic pigs [24].

The slow growth rate of wild boars has prompted farmers to cross them with domestic pigs [77] to obtain a higher final live weight [78] as well as less aggressive animals [73,79]. The growth rates reported for 2n = 37 and 2n = 38 crossbreeds were close to results reported in the literature for crosses between domestic pigs and wild boars (330 to 470 g day^−1^) [76,80].

Wild boars, like all wild ungulates, are capable of providing satisfactory slaughter yields with well-developed bacon and shoulder cuts but which would be less acceptable than those of domestic pigs due to greater contributions of the head and hard skin, which is densely covered with thick black bristles [24,31].

Differences in dressed weights (kg), assumed to include heads and skins but without thoracic and abdominal organs, were found in wild boars hunted in different European countries due to animal age (7–12 months, 24.6–30.8 kg; 13–24 months, 53.0–64.9 kg) and gender (>24 months old, 87.2–103.8 kg for males and 66.3–84.2 kg for females) [24]. In agreement with these results, different dressed weights have been recorded for males (65–108 kg) and females (50–80 kg) aged 3–4 years, hunted in Romania [24].

No differences were found for carcass weight (CW) and backfat thickness (BT) among Chilean purebred wild boars (male and female) fed 20% (*w*/*w*) acorns in their diet or a commercial acorn-free diet [32]. The differences were significant for wild boars eating 40% acorns in their diet, and the values were 37.6, 40.1, and 44.6 kg for CW and 11.8, 14.0, and 15.6 mm for BT for wild boars consuming a commercial diet, 20% and 40% acorn diets, respectively.

Summarizing the data published in Polish by different authors [24], the carcass yields of wild boars harvested in Poland ranged between 59.9 and 74.3% (skin contributed 15.7–29.4% of the initial weight) and increased with body weight. Comparable values for piglets (<1 year; 75.1–77.4%), yearlings (1–2 years; 73.7–79.6%), sub-adults (2–3 years; 74.9–80.1%), and adults (63.2–81.9%) were reported for wild boars hunted in Croatia and for wild boar × wild pig crosses (76.5–78.6%) hunted in the United States of America [81].

Higher carcass yields (82.8–84.0%) were instead reported for wild boars of medium live weight hunted in Italy [31]. The carcass characteristics of wild boars were almost always influenced by the age of the animals and other variables through the interaction with sex [31]. Head and hind leg (as a proportion of half carcass weight) of wild boars hunted in Europe (Karpaty and Maremma subspecies) decreased with animal age (8.48/24.93, 7.40/23.98, and 7.75/23.78%, head/hind leg in animals aged 7, 9, and 12 months, respectively). In contrast, loins and steaks increased with increasing age of the animals (7.37/12.37, 7.61/14.82, and 9.26/15.08% in 7, 9, and 12 months, respectively) [31].

Greater liver and heart weights were reported in wild boars compared to domestic pigs (Pietrain, a selection result for high meat production) and in their crossbreeds, after the same fattening period of 210 days [78]. This is related to the physiological states of the animals in the wild, which are different from those of domesticated animals. In wild boar × Pietrain crosses, the F1 and F2 hybrids gave lower meat and higher fat yields than the average of the parent breeds [24,78].

Studies have indicated that the adipose tissue and loin area increased in crossbreeds with an increasing proportion of wild boar heredity, while other authors reported lower ham weights [24,77]. These differing findings could likely be related to differences in chromosomal traits that code for the composition and/or proportion of body parts [80].

### 5.2. Meat Quality

The literature has outlined microstructural differences in wild boar meat compared to domestic pork meat. In regard to the proportions of the different types of muscle fibers, which are related to the physical, chemical, and morphological characteristics of the meat [82], muscles from wild boars show a higher proportion of slow oxidative (I) fibers compared to domestic pigs and this has been explained by the behaviors of wild boars in the wild involved in foraging for food with attention to predators [24].

Skewes et al. (2014) [76] compared the fiber characteristics of *M. longissimus dorsi* (LD) and *M. semimembranosus* (SM) of European wild boars, which are phenotypically similar but differ in karyotypes (2n = 36, 37 or 38). Differences in the distribution of type IIA fibers in LD muscle were observed, being higher in the 2n = 37 than in the 2n = 38 karyotype. An intermediate proportion was detected for the 2n = 36 karyotype. No differences due to chromosome number were found between the groups for the other fiber types, i.e., I, IIB, IIB oxidative, and IIB non-oxidative. Differences in fiber proportions between SM and LD muscles were also noted. More intense red meat color was found for karyotype 2n = 36 than for 2n = 37 and 2n = 38. Moreover, there was a smaller muscle fiber cross-sectional area in the 2n = 36 karyotype than in the 2n = 38, and this could be related to the darker color of the wild boar meat. It has been argued that muscle fibers are modified continuously during the life of the animal due to different circumstances; the higher percentage of red, slow-twitch, oxidative (I) fibers in wild boar meat could be related to the animals’ increased exercise [76,83]. The information obtained from the study could be useful in improving the quality of wild boar meat.

Furthermore, wild boar meat has a darker color than that of swine, stronger muscles, lean meat, and in general, is considered less juicy and tender [24,61].

Flavor, often referred to as a “game meat” flavor, is another crucial element distinguishing pork from wild boar meat [84]. Lammers et al. (2009) [85] reported that the typical aroma of wild boar meat could be due to Maillard reactions followed by Strecker degradations of the sulfur-containing essential amino acid methionine and phenylacetaldehyde (derived from phenylalanine). However, the castration of males, before reaching sexual maturity, prevents the meat from taking on the characteristic flavor [31]. In females, on the other hand, this problem is not present because the meat of non-ovariectomized subjects, as occurs in the swine species, possesses organoleptic properties that are appreciated by the consumer [31]. Castration would be necessary only if the production cycle extends beyond the puberty age of the animals to reach heavier final live weights [31]. The purchase of this type of meat is highly conditioned by its sensory properties, among which, tenderness is a very important requirement for consumers. Sensory evaluations attribute higher average scores for flavor (intensity and desirability) and taste (intensity, desirability, juiciness, and tenderness) to domestic pig meat and crossbreeds compared to wild boar meat [24]. However, recent studies found a different acceptability and an increasing demand for wild boar meat by consumers, due to its healthiness, specific sensory profile, and low environmental impact [5,86,87]. A sensory evaluation has recently judged that wild boars fed with acorns produced juicier and more tender meat [32].

A greater initial pH (>6) at 45 min (pH_45_) post-mortem is measured in wild boar meat compared to that in the meat of European domestic breeds (<6) [78]. The authors also reported a greater pH at 24 h (pH_24_) post-mortem in wild boars than in domestic pigs [24,31]. In contrast, pH values measured in the medial part of the hindlimb region of wild boars hunted in the Western Italian Alps, between 30 min and 6 h after killing, did not differ from those obtained in domestic species [50]. Furthermore, when subsequently comparing the pH of the *M. longissimus dorsi* in carcasses of Brazilian commercial wild boars and domestic pigs stored at 0–2 °C, the values decreased more slowly in wild boars than in pigs [24]. This would indicate a greater resistance to stress in wild animals compared to domestic ones, and has been related to the higher proportion of slow-twitch fibers (I) and lower proportion of fast-twitch fibers (IIB) in the *M. longissimus dorsi* of wild boars compared to pigs [24]. It has been remarked that higher contents of oxidative fibers (I) slows the rapid postmortem glycolysis and the development of the pale, soft, exudative (PSE) condition in meat [88]. Furthermore, comparative studies of boar, pork, and crossbreed meat revealed lower drip losses for wild boar muscles than those of pigs and crossbreeds [5,24,31].

The average chemical composition of wild boar meat shows high levels of proteins (22–26%) and low total lipid contents (2–5%), which are composed of oleic (36–43%), linoleic (13–19%), linolenic (0.6–1%), and palmitic (20–21%) acids [24,83,84]. The lipid and cholesterol levels (55–59 mg/100 g meat) are lower than meats from other ruminants [24,89,90]. The cholesterol content of meat from animals with different karyotypes (2n = 36, 37, and 38) reported in the study by Skewes et al. (2009) [73] varied between 20.9 and 37.2 mg/100 g and between 34.4 and 36.9 mg/100 g, for the semimembranosus (SM) and longissimus dorsi (LD) muscles, respectively. There were no significant differences in the meat cholesterol content due to chromosome number. However, higher percentages of polyunsaturated fatty acids (27.8, 13.17, and 7.53% for 2n = 36, 37, and 38, respectively) and lower C16:0 (20.87, 23.02, and 23.78% for 2n = 36, 37, and 38, respectively) and C18:0 (9.59, 15.12, and 16.85% for 2n = 36, 37, and 38, respectively) were reported in SM muscles from wild boars (2n = 36) compared to crossbreeds (2n = 37, 38). On the other hand, karyotype did not affect the fatty acid composition of LD muscles.

Flores Ahumada et al. (2021) [32] found that purebred Chilean wild boars (male and female) consuming acorns (20 and 40% of diet, *w*/*w*) had lower percentages (23.11 and 23.51% with 20 and 40% acorns, respectively) of C 16:0 in longissimus lumborum (LL) muscles than those consuming a commercial acorn-free diet (24.56%). In contrast, the proportions of monounsaturated (MUFAs) and polyunsaturated fatty acids (PUFAs) were higher in LL muscles from wild boars fed with acorns (48.03 and 48.11% and 14.43 and 14.47% for MUFA and PUFA, and for 20 and 40% acorns, respectively) than those found in animals fed only the commercial diet (47.3 and 14.06% for MUFA and PUFA, respectively). Also, the meat cholesterol contents were lower in wild boars fed acorns (21.6 and 22.2 mg/100 g meat, for 20 and 40% acorn diet, respectively) than in meat from animals fed a diet without acorns (23.9 mg/100 g meat). This study highlights the importance of acorns in wild boars’ diets in the quality of wild boar meat products.

Overall, the protein content of wild boar meat is similar to that of domestic animals such as chicken (20%), sheep (20.27%), and rabbit (22.23%) [31]. Differences in intramuscular fat content correlated with differences in food availability between seasons and/or estrus, which could influence food intake [24].

Table 1 (using data from [90,91,92]) compares the chemical compositions of the longissimus dorsi muscle of wild boars hunted in Italy (WB), reared wild boar (RWB), wild boar × domestic pig crosses (WBDP), and domestic pigs (DP). Marsico et al. (2007) [90] found lower moisture and fat (ether extracted) contents and higher protein concentrations in meat from WB compared to RWB, WBDP, or DP. Moreover, WB meat had a higher percentage of n − 3 fatty acids (2.9%) than animals reared on complete diets (RWB, WBDP, and DP; 1.3, 1.28, and 1.03%, respectively).

Diversity in eating behavior or differences in food availability between different habitats were related to variable contents of oleic and linoleic acids in *M. psoas major* from wild boars hunted in Portugal [89]. The percentages of the linoleic acid isomer C18:2 *cis*–9 *trans*–11 (conjugated linoleic acid, CLA) were also greater in wild boar meat than in pig meat, most likely deriving from the biosynthesis of intestinal bacteria [93].

Considering that the ratio between polyunsaturated and saturated fatty acids (PUFAs/SFAs) in the meat of wild boars from Portugal (0.52–0.60) or Lithuania (0.43–0.53) was above the minimum ratio of 0.40, this meat may contribute to a reduction in the risk of cardiovascular diseases in humans [89,94]. Furthermore, the atherogenic and thrombogenic indices, indicative of the risk of cardiovascular disease, decreased in the meat of wild animals hunted in Italy (WB) vs. commercially farmed meat (RWB, WBDP, DP) [90]. Weight, sex, or month of hunting did not affect these indices. Factors such as the live weight of animals, sex, and hunting season and region could have a great influence on the meat quality of wild boar, in particular on fatty acid profiles [94,95].

The macro and micro mineral contents in muscles of wild boar are often correlated to differences in soil mineral concentrations in the areas where the animals were hunted [96]. For example, different contents of calcium, phosphorus, and zinc in the meat of Hungarian wild boars, living in parks where different feeding programs were applied (natural diet, supplementary feed, complete commercial diet), were correlated with differences in the mineral content of the park soils [96]. However, the manganese and selenium contents, even with different concentrations in the park soils, did not vary [24]. Recently, there were no differences between Cd and Pb contents of meat from wild boar with different live weights and sexes, hunted in two different areas of the Molise region and those reported in the literature (ranging from 0.001 to 0.355 mg/kg for Cd and ranging from 0.03 to 0.441 mg/kg for Pb) [95,97].

In regard to vitamins, Quaresma et al. (2011) [89] found that in *M. psoas major* from wild boars hunted in Portugal, 71% of the total isomers of vitamin E was represented by α-tocopherol, which increased in adult males (19.2 μg/g meat) and in females (18.1 μg/g meat) compared to younger animals (15.5 μg/g meat). Quaresma et al. (2011) reported a greater α-tocopherol concentration in wild boar meat than the concentrations (15.1–16.3 μg/g) that had been, previously reported [98] in *M. psoas major* of pigs fed high levels of dietary vitamin E (700 mg/kg feed). Recently, Palazzo et al. (2021) [95] found that the α-tocopherol concentrations of longissimus thoracis muscles from wild boars hunted in Italy were two-fold lower than those reported by Quaresma et al. (2011) [89], ranging from 5.87 to 6.05 μg/g meat, without any influence of the live weight or sex of the animals. These differences could mainly be attributed to the genotype and types of muscle, which have different vitamin needs in relation to the functional and metabolic differences of their fibers and different fat contents [99]. It has been reported that higher α-tocopherol concentrations could result in a longer shelf life and a delay in the onset of discoloration in wild boar meat compared to pig meat, due to the increased protection of lipids and myoglobin from oxidation [24].

### 5.3. Processed Products

Paleari et al. (2003) [100] stated that if wild boar meat is used without being transformed, it would be unsuitable for large-scale domestic consumption, as it is not easy to cook, and sometimes not easy to sell, even at a low price. Wild boar meat could be conveniently used in processed products (sausages and cooked and/or raw hams), whose ripening would also be favored by an optimal muscle acidification process [24,31]. In dry-cured fermented sausages, textural differences have been mainly related to pH, which explains the variability among different brands of Chorizo de Pamplona. It has been reported that pH evolution during the ripening process strongly affected texture changes and the possibility of obtaining a more consistent product [101,102,103].

Moreover, wild boar meat would be characterized by a greater water-retention capacity than that of domestic pigs, and this would also make it particularly suitable for use in processed products [24,31]. Wild boar ham was also confirmed to be tougher and require more masticating in comparison with pork ham [61].

Massaging wild boar muscles was a technique to test the efficacy of mechanical tenderization for conditioning the meat. Fatter (and therefore structurally more modifiable muscles) showed different rheological values with massage application. However, older animals had less structurally modifiable muscles than younger animals [24,82]. Cold curing of muscles under vacuum, for a 7-day period at 4 °C, with or without marinating agents (red wine, calcium chloride, pineapple juice, and kefir), increased the tenderness and juiciness [82].

The use of the shoulder instead of the hind legs in a traditional mold-ripened salami made from wild boar meat [104] slightly lowered the protein content but significantly increased the concentrations of hydroxyproline and biogenic amines (histamine, putrescine, and cadaverine) at day 35 of fermentation and drying. The higher content of unsaturated fatty acids and the use of wild boar adipose tissue but not starter bacterial cultures, likely increased the salami TBARS (thiobarbituric acid reactive substances, a common measure of lipid peroxidation products) content by 2–3 times compared to pork back fat. Based on the odor and taste, the salami with pork back fat and bacterial starters was preferred by a semi-trained sensory panel.

Proteolysis indices (non-protein nitrogen/total nitrogen × 100) for saucissons elaborated using wild boar meat were greater (29.9–30.8%) compared to values reported for the same type of processed products prepared with pork or beef (12–20%). The differences were related to different microorganism proteolysis processes during maturation. Furthermore, the authors argued that contamination of wild animal carcasses in the field by microorganisms is higher compared to that in domestic animal carcasses [105].

Instrumental texture parameters (hardness, chewiness, springiness, and cohesiveness) at an internal temperature of 68 °C were higher for cooked wild boar sausages compared to sausages made from pork or beef shoulders [24]. A sensory evaluation confirmed the instrumental results, judging wild boar sausages to be less juicy and gummy, and more springy, than pork ones. Sausages made from the meat of wild boars shot in the autumn–winter season had higher textural values and were juicier compared to wild boar shot in the spring–summer season. However, safety (biogenic amines, microbial load, and polyamines) and quality requirements (physicochemical characteristics, sensorial properties) of a spreadable, raw, processed product elaborated with 60% meat from wild boar shoulder, 40% pork back fat, and microbial starters, were similar to those expected to be obtained with the use of pork only [24].

Paleari et al. (2003) [100] evaluated the physicochemical characteristics of different cured, fermented products, elaborated utilizing different types of meat (beef, horse, wild boar, deer, and goat) but the same manufacturing process similar to that of bresaola. In terms of physicochemical parameters, wild boar end products had lower values of pH (6.30), *a*_w_ (0.90), moisture (48.2%), and saturated fatty acids (SFAs, 35.5%) than beef end products (6.72, 0.95, 55.4%, and 47.8% for pH, *a*_w_, moisture, and SFA, respectively). Protein content (39.3%), total free amino acids (2315 mg/100 g), polyunsaturated fatty acids (PUFAs, 16.2%), and monounsaturated fatty acids (MUFAs, 45.7%) were greater in wild boar meat products compared to those with beef (34.6%, 1338 mg/100 g, 6.5, and 43.6%, for protein content, total free amino acids, PUFAs, and MUFAs, respectively).

Different pork ham cooking treatments are clearly capable of altering pork meat consumption and sensory perception due to differences in juiciness, fibrousness, and tenderness, which are of great importance for meat quality acceptance [61,106].

Ilic et al. (2022) [61] examined the impact of boiling, grilling, and sous-vide cooking methods on wild boar meat’s textural, oral processing, and sensory qualities. The applied cooking methods affected wild boar texture parameters such as hardness, chewiness, springiness, and cohesiveness, but the wild boar flavor dominance rate remained persistent for all three cooking methods. The used cooking methods similarly affected the dynamic perceptions of firmness, fibrousness, juiciness, and flavor, as in the case of pork ham. The study authors concluded that cooking methods are promising tools for tailoring game meat for consumption and eating experience.

Recently, Freschi et al. (2023) [14] investigated the sensory attributes for ten types of “cacciatore” salamis, one of the most widespread types of salami in Italy, prepared with different mixtures of wild boar/pork (30/50 or 50/50) and spice ingredients. The main findings of the hedonic test revealed that the flavorings used received the highest scores, as well as satisfactory acceptance, regardless of the ratios of wild boar to pork in the salamis. According to the authors, doughs with a high proportion of wild boar meat might be used without affecting product preference and allowing the production of more cost-effective and environmentally friendly products.

## 6. Discussion

### 6.1. Scientific Opinion on Wild Boar Management

In order to improve the management of ungulate populations, several authors have proposed implementing an adaptive management approach, depending on the situations (contexts) of overabundance [19,20]. In Europe, wild boar *Sus scrofa* was found to be overabundant in arable farming, livestock farming, and peri-urban areas, and indicators of ecological change have been proposed to monitor this population [19].

Rational wildlife management proposals, also with a view to safeguard biodiversity, include the protection of predators, in order to restore their natural role in controlling the wild population [107].

In contrast, the transfer of populations to areas other than their native ones and wild–domestic hybridization, especially in recent years, have also become extremely necessary to ward off the danger associated with the spread of diseases such as classical swine fever, African Swine Fever (ASF), and animal tuberculosis (TB) [19], with serious economic repercussions for the global pig industry [12,13]. It has been reported that wild ungulate populations in general, and wild boar in particular, pose a significant threat to sympatric outdoors pigs through the transmission of diseases, such as TB in Mediterranean Europe or ASF in the central and eastern regions of Europe [13].

In Europe, the consumption of wild boar meat comes mainly from hunting game estates, while farming systems that include grasslands for grazing are prevalent in countries such as the United Kingdom or Finland [71]. Wild boar populations have increased significantly in the late 20th century, and that was largely associated with widespread supplementary feeding, reforestation, intensification of agricultural activities, and mild winters [71].

In Europe, breeding areas are frequented by wild ungulates and agricultural and hunting activities largely overlap [19,108]. The number of free-range animals has significantly decreased in the last twenty years in Europe [19] and this has changed the land use, favoring natural vegetation and conditions more suitable for recolonization by wild ungulates [109]. However, under limiting conditions (summers in the Mediterranean and winters in northern Europe), food shortages for ungulates may occur [19]. More studies are needed to evaluate how certain management actions (e.g., supplementary feeding, hunting, and fencing) may affect ungulate population density [19]. The primary objective of winter feeding of ungulates in Europe is the prevention of environmental damage, particularly to commercial and native forests [110]. However, Milner et al. (2014) and Putman and Staines (2004) [56,59] reported limited evidence of the effectiveness of supplemental feeding to protect forest and natural habitats, and any positive effects were often undermined by increased ungulate population densities. Therefore, supplementary feeding has the potential to alter population dynamics, leading to an overabundance situation in some locations [19].

To reduce the risk of pathogen transmissions at the wildlife–livestock interface, a number of management actions, such as movement restrictions, controlling wild ungulate populations through culling and/or improving farming practices, removal of harvested animals, translocation, barriers, deterrents, arthropod vector control, vaccination of wildlife, and can be used in relation to the target pathogen [111]. A detailed standardized protocol evaluating and implementing farm-specific preventive actions against wildlife interactions was recently developed for the first time in extensive pig production systems [13]. In the current context of the diffusion of ASF across Europe and the threat posed by many other shared diseases, protocols that can be adapted or extrapolated to be incorporated into control and eradication strategies are in high demand [13]. Nonetheless, even though fencing tools can limit wild boar movements, the presence of streams and other points of vulnerability would not make them 100% effective [13].

### 6.2. Potential of Wild Boar Meat

As pork is produced in a great variety of systems, there is not just one type of solution for the management of wild boar, and measures should, therefore, be adapted to the different environmental situations [4,14,112,113].

A sustainable farming system requires knowledge of all dimensions of sustainability that commonly include the economy, environment, and social wellbeing [2,4]. Figure 2 (adapted from [2]) shows a typical sustainability model, the so-called triple P model.

There are three overlapping ellipses which reflect the social (people), economic (profit), and ecological (planet) dimensions. As highlighted by the author [2], only the intersection of all three can be regarded as sustainable, although overlapping only two dimensions might be viable, bearable, or equitable. Animal health and welfare (AHW) is often included in the social dimension, although due to the growing interest of society in this regard, it should be a separate fourth dimension. All dimensions are strongly interconnected and must all be considered when making business management decisions [2,4].

The exploitation of wild animal species, in addition to providing quality meat, could also allow a wider and more rational use of marginalized territories, and hilly and mountainous regions [5,31,65]. The productive recovery of these areas can be used to pursue food production purposes, as well as wildlife hunting, agritourism, and educational purposes by integrating the income of companies with marginal productivity [31,54]. Furthermore, protection and restoration of endangered and threatened species can enhance areas unfavorable to crops and meet recreation and leisure needs [31,54]. This type of breeding, therefore, would not replace that of domestic livestock but would represent an integration to traditional zootechnics or other forms of land management, including forestry [19,31]. Wild boar meat has a negligible carbon footprint compared to meat produced from industrialized farming, as it is a natural product [5]. Accurate data on greenhouse gas emissions produced by 1 kg of wild boar meat are not available; the methodology for the exact detection and measurements could be an independent research topic [5].

There is limited research on the effects of hunting methods, carcass dressing, and treatment on meat hygiene, yield, and quality [24,65]. Furthermore, the effects of age, sex, and season of shooting on the carcass composition and chemical characteristics of wild boar meat are less studied [65].

Despite the nutritional benefits of consuming wild boar meat, proper hygienic, and technological controls of in situ management are needed to reduce the probability for the occurrence of zoonoses and food poisoning, including infectious diseases caused by bacterial pathogens (e.g., *Yersinia enterocolitica*, *Brucella suis*, *Salmonella* spp., *Leptospira* spp., and *Escherichia coli*) and endoparassitosis (e.g., *Trichinella* spp.) [5,24,65]. Requirements regarding the hygienic quality of game meat, the risks of zoonoses, and safety standards have been established in the European Union (EC, No. 853/2004) [95]. The risk of infection with *Trichinella* is well known and to this purpose, specific controls are required by Directive (EC) No. 1992/45 of the European Union [65]. Following an outbreak of trichinosis in Ontario, Canada, it was suggested that wild boar meat should be cooked until all meat reaches a temperature of 77 °C [24].

The microbiological quality of game meat is affected by many factors such as microorganisms on the skin, digestive tract, and muscles, the situations in which the animals are killed, and the subsequent slaughter conditions [65]. Microflora that develops during the storage of meat is related to the storage conditions and intrinsic biochemical qualities of the meat [114]. Given the variability of conditions, the microbiological quality of ungulate meats has also been found to be highly variable [65]. However, it has been reported that if animals are correctly shot and properly dressed, microbial contamination of fresh carcasses may be very low [24,65]. There are no specific microbiological criteria for game meat within European Union legislation, since the microbiological quality of wild boar meat is considered to be similar to that of domestic pigs [24,65]. Given the low prevalence of pathological agents in wild boar meat, it was argued that future research should focus on factors affecting spoilage and shelf life of meat rather than safety issues [24,50]. There is a lack of studies on wild boar hunting practices that could improve meat yield and quality [65].

Little information is also available on heavy metals, radionuclide, organochlorine pesticide, and polychlorinated biphenyl contamination of meats from wild ungulates [65]. It has been reported that wild animals, and especially wild boar, are good bioindicators of environmental pollution such as heavy metals (cadmium, Cd, mercury, Hg, and lead, Pb), radioactive isotopes released into the environment (e.g., radiocesium after the Chernobyl nuclear accident), pesticides and biphenyls containing chlorine, and synthetic organic substances which are used in agriculture [95,115].

The safety requirements of game meats are addressed by Regulations (EC) No. 853/2004 [95]. However, there are no specific limits regarding the concentration of heavy metals that are known to be harmful to animal and human health in meat, such as for meat and offal of farm animals, as stated by Regulation (EC) 1881/2006 and the later amendment Commission Regulation (EC) 629/2008 for the meat of farm animals (bovine animals, sheep, pigs, and poultry) [95]. It has been reported that sampling may affect the heavy metal contents of wild boar meat, i.e., muscle samples to be analyzed from parts of the body compromised by the shot [24]. In particular, differences in Pb levels in muscle could be related to lead dispersion in the animal body due to bullet fragmentation and this topic has received great attention in the literature due to the impact on human health [95,97]. The liver and kidneys are generally the main sites of heavy metal bioaccumulation, followed by muscle and fat [24,95].

However, since the accumulation of individual contaminants shows regional variability, which is related to local pollution sources, coordinated investigations at the national and international levels are recommended [65]. These studies are of fundamental importance in establishing the market value of meat and stimulating new food technologies [24,61,63,65].

Moreover, as shown in farmed animals, stress experienced by wild animals before death has very important consequences on meat quality and an accurate placement of the shot should achieve rapid death while minimizing suffering and avoiding carcass contamination [65]. Furthermore, “good rules” in hunting practices include avoiding burying or improperly dispersing offal and scraps in the environment as well as reporting any cases of wild boar carcasses on which the health services can carry out adequate checks [116,117,118]. In the framework of Regulation (EC) No 852/2004 (European Commission, 2004), the skinning and butchering should be performed at “approved game handling establishments”, under veterinary control and with formal Hazard Analysis and Critical Control Point (HACCP) principles [65].

The importance of including farmers in the study of appropriate awareness campaigns on this topic was highlighted in [13,65]. This would allow the early identification of any dangerous outbreaks of infection for humans and other animals, allowing rapid and decisive interventions. Moreover, in order to pursue these goals, there is a need for hunters and expert figures to be able to determine the number, sex, and age of the animals to kill, based on the structure [20,65] of the population in a given area.

The different wild forms of *Sus scrofa* (wild boar, hybrid, and feral pig) need investigation and study to determine their production potential, not only with the aim of producing high-quality meat, but also to adjust its production seasonally to supply the market throughout the year [61]. To this end, Hodgkinson et al. (2017) [119] carried out interesting studies comparing European wild boars and domestic pigs. The authors compared growing European wild boars (60 to 207 days of age) and domestic pigs (Landrace × Large White) in a semi-extensive management system, in order to detect grazing grass intake along with behavior. The grazing consumption (by metabolic body weight) of European wild boar was higher than that of domestic pigs. Approximately 0.20 of the total dry matter (DM) intake by wild boar was grazed, while it was only 0.10 in domestic pigs. However, domesticated pigs ingested a larger amount of supplementary feed than wild boars. Furthermore, grazing provided 0.20 of the total apparent daily energy consumption in wild boars, compared with 0.07 in domestic pigs. Regarding behaviors, during the 8 h grazing, wild boars spent 0.54 of their time grazing or moving, resulting in more activity, compared to 0.32 of the time for domestic pigs.

It was previously observed [120] that the European wild boar exploits 20% less digestible energy (8.48 vs. 10.56 MJ DE kg^−1^ DM) from fiber-rich foods (alfalfa meal with 50.4% NDF) compared to the domestic pig (Landrace × Large White). It was concluded that for ingredients containing relatively low concentrations of fiber (such as corn and oats), the DE values determined in domestic pigs can be validly applied for wild boar diet formulation.

In another study [67], it was estimated that wild boars would have met slightly less than 90 and 45% of their daily maintenance energy requirements as DE by grazing on paddocks of *Lolium perenne* in spring and summer, respectively. Individual forage consumption ranged, on average, from 210 to 550 g of DM per day depending on the season and the predominant forage species. DM, energy, crude protein, and amino acid intakes varied greatly between days, but did not differ significantly in the amount between paddocks of *Lolium perenne* and *Plantago lanceolata*. A significant correlation was reported (r = 0.59, *p* < 0.0001) between the availability of DM and consumption. The correlation between DM consumption and mean daily temperature was not significant (r = 0.05 in spring and r = −0.07 in summer, *p* > 0.05) [67].

The observation of grazing consumption and behavior of European wild boars over a five-day period showed no differences between continuous and rotary grazing systems [121].

Rivero et al. (2013) [122] investigated the effect of grass availability on forage DM consumption by ring-snouted boars and reported that grazing consumption can be increased by increasing grass availability and greater weight gains can be achieved in wild boars with access to pasture compared to those without grazing access. In their study, forage did not limit growth or feed conversion efficiency. The authors concluded that a high percentage of consumed DM, even 26%, can be provided as forage instead supplemental diet, which can increase body weight gain.

The large individual variations in pasture consumption [67] indicate great genetic variability in *S. scrofa* L. that could be exploited through breeding programs [123].

However, there is no information on the amino acid content of fresh grass that is digestible by wild boars [71]. In wild boar meat farming systems, particularly in the intensive ones, the definition of protein and energy requirements of these animals has not been determined and therefore, there are no feeding programs capable of making the breeding of this ungulate profitable and responsive to the market needs [31,71]. High quality is attributed along with a high price to meat and end products obtained from free-living wild boars that have consumed acorns, but there are no studies on the effects of acorns or combinations of commercial foods and acorns on the meat quality of wild boars in confinement [32].

The use of low-cost agro-industrial wastes as part of wildlife feeding [63] can be seen as a strategy to reduce the production cost of wild boar.

The most beneficial property of wild boar meat is its low intramuscular fat content with a favorable acid profile, even though it is darker and less tender compared to pork. However, even though there are distinctions between pork and wild boar meat that may cause different consumption and sensory perceptions, research on these aspects is still lacking. Furthermore, although the composition of muscle fibers is markedly different in wild boar and pig muscles, with important effects on meat quality, the physical characteristics related to these different fiber distributions are poorly studied. [24]. Food technology studies are needed to expand our knowledge on how wild boar meat is best consumed and perceived in order to stimulate the development and establishment of new end products [61,63]. Research is needed on the impact of preparation methods on wild boar meat’s textural, oral processing, and sensory qualities [61].

Furthermore, the development and application of molecular genetic tools to differentiate different species (wild boar and domestic pig) and to verify the regional provenance of the meat would be desirable to certify local production chains [65]. A premium price is obtained when consumers positively consider the link between the animal used and the perceived quality of its products [124]. It is essential to develop adequate pre-slaughter [125] and post-mortem processing procedures that confer distinctive and high-value characteristics to the final product [54,126,127].

For this purpose, without direct market demand (and by paying farmers more for produce), only a government law can give the animal industry an incentive to change in the face of an unfavorable economy [15]. A law or market action is needed for any sustainability issue to trump the economy. Science can be used to identify more sustainable systems of pig farming and pork production, but it must be viewed in the context of human emotions and economics to obtain a sense of which systems and practices are the most sustainable [15].

## 7. Conclusions

The literature suggests the potential of farming wild boar but there is no single solution to raising this species for meat production. Throughout the world, wild boar is mainly raised using free-range systems, which take advantage of the wild boar’s ability to explore the different foods available. Since the One Health approach requires the development of models for the integration between humans, animals, and the environment, the assumption that pigs’ foraging through grazing contributes to the development of more natural production systems that are respectful of the environment as well as cheaper. Semi-extensive farming systems would allow animals to have more natural behaviors than those of intensive farming systems. Therefore, grazing-based production systems may be one way to achieve long-term sustainability of wild boar production. However, continuous research on their performance in different production systems, including intensive ones, is needed to achieve the economic, social, and environmental sustainability of farming. Furthermore, studies concerning the quality and acceptability of processed products are needed, as this type of meat could potentially be used in different meat end products.

## Figures and Tables

**Figure 1 animals-13-02258-f001:**
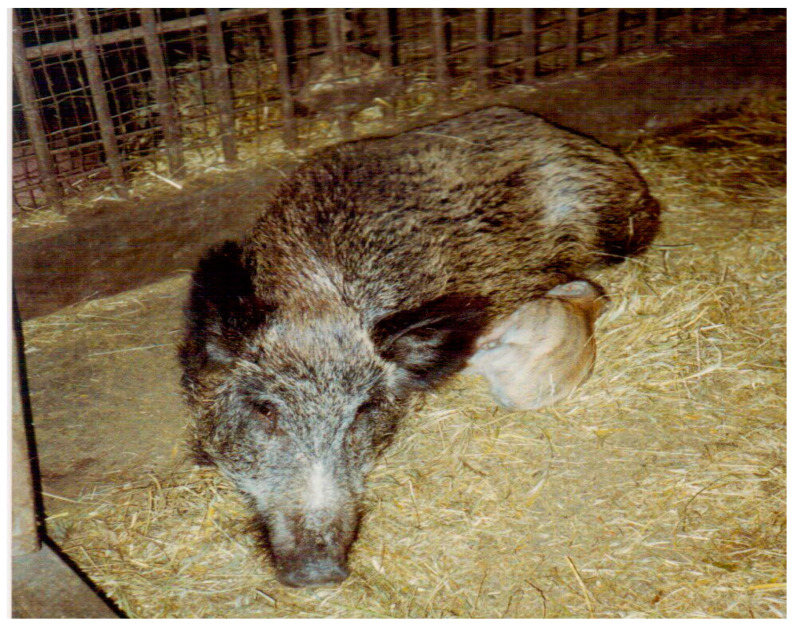
Female wild boar, Karpaty subspecies, with a piglet in a semi-intensive breeding system.

**Figure 2 animals-13-02258-f002:**
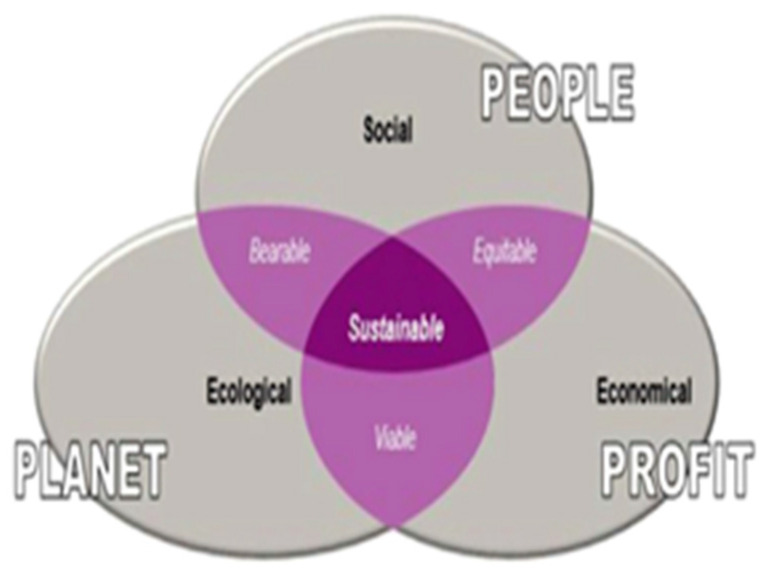
Three-dimensional model of sustainability.

**Table 1 animals-13-02258-t001:** Physicochemical measures of *M. longissimus dorsi* of wild boars hunted in Italy (WB), reared wild boar (RWB), wild boar × domestic pig crosses (WBDP), and domestic pigs (DP) [90,91,92].

Characteristic	WB (n = 4)	RWB (n = 4)	WBDP (n = 4)	DP (n = 4)	SED ^a^
Physical					
pH_45_	6.35	6.41	6.61	6.04	0.251
pH_24_	5.48	5.94	5.74	5.49	0.193
L* (lightness) ^b^	43.62	45.92	47.85	50.42	2.591
a* (redness)	12.39	7.26	6.37	5.28	0.968
b* (yellowness)	11.97	10.64	10.23	9.61	1.539
WBS (kg/cm^2^) ^c^	2.42	2.54	2.85	4.39	1.041
Cooking loss (%)	31.22	18.52	14.96	11.86	3.476
Chemical (%)					
Moisture	70.50	73.41	73.65	71.37	1.367
Protein	25.87	22.50	22.24	21.35	0.893
Fat	1.55	2.00	2.15	4.56	1.010
Ash	1.23	1.30	1.27	0.86	0.127

^a^ Standard error of difference. ^b^ Color evaluated according to the CIELAB L*a*b* scale [91]. ^c^ Warner–Bratzler shear force values of samples cooked in microwave to internal temperature of 75 °C [92].

## Data Availability

Not applicable.

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
