# Peer review of "Use of Wild Boar (*Sus scrofa*) as a Sustainable Alternative in Pork Production"

_animals, 2023, doi:10.3390/ani13142258_

Round 1

Reviewer 1 Report

This paper could be of interest for animals. However there are several points that need to be revised:
I think that the wild boar is an interesting alternative for pork production, on this context, several issues  were included in paper, but other important aspects were out from the review. The wild boar is a interesting alternative to produce meat in outdoor conditions, a lot information about there in US, Europe and Latin-American. Please, include this information to improve the manuscript. Information of effect from different type of feed (ej., acorns, pasture and branches) on animal nutrition and meat quality should be included in order to add discussion of the wild boar production sustainability.  

Other issue important to increase wild boar productivity is crossbreed with pigs, because increase of live weight and rising rate (Muller et al., 2001; Skewes, et al 2008). I encourage to author to review this aspects in profundity that to increase the input of this paper to knowledge of this issue.

Please intent to avoid cite in excess from other reviews (ej. Sales & ; Kotrba, 2013) , please go direct to result manuscript

Line 311: Add studies of effect of karyotype animals of muscle fibre (Skewes et al., 2014 http://dx.doi.org/10.1016/j.meatsci.2014.06.001)

Line 354: Please review studies (Skewes et al., 2009 doi:10.1016/j.meatsci.2009.04.017)

Line 375: Why there is CLA in wild boar because this fatty acids is only form ruminants, Please include possible reasons from this finding.

Line 376: Please, review studies Flores et al., 2021  t t p s : / / r e v i s t a s . r e d u c . e d u . c u / i n d e x . p h p / r p a / a r t i c l e / v i e w/ e 3 6 0 8

Line 517: Please review the manuscript form Dra. Hodgkinson https://www.researchgate.net/profile/Suzanne-Hodgkinson

Line 545-471: please eliminate this paragraph, this is redundant and don’t add value to manuscript .

Line 657-660 Please review the manuscript form Dra. Hodgkinson https://www.researchgate.net/profile/Suzanne-Hodgkinson

Conclusion: should be re-written according to discussion suggestions.

Author Response

Attached reply as a file

Reviewer 2 Report

The manuscript titled "Use of wild boar (Sus scrofa) as a sustainable and alternative pork production" based on a sustainability and "One Health" perspective, integrates various literature sources to highlight the problems of intensive farming processes. It then introduces the background of wild boars, compares the different systems and methods of hunting wild boars in various countries, and provides a detailed introduction to the carcass characteristics and meat quality of wild boars, suggesting that they could be a potential source of pork. Overall, the manuscript presents a novel perspective and discusses the possibility of "wild boar to the table" from the perspectives of sustainability, management, and nutrition, aligning with the requirements and scope of this special issue and providing an interesting supplement to traditional animal husbandry. However, further refinement is needed to enrich the manuscript:

lLine-39:It is necessary to add further to the concept of "One Health". lLine-423:I am skeptical of the notion that "wild boar meat would be characterized by a greater water holding capacity than that of domestic pigs." Generally speaking, wild pork usually contains more muscle fibers and connective tissue, which makes the muscle more elastic, so the water holding capacity is lower. Table 1 also shows that wild pork has a higher proportion of cooking loss. lWhat readers may be concerned/interested in:

1. The number of wild boar and the status of wild boar in different countries or regions.

2. Sales and prices of wild boar meat in different countries or regions.

3. Policies on hunting/trafficking of wild boar meat in different countries or regions.

Author Response

Attached reply as a file

Round 2

Reviewer 1 Report

The manuscript incorporated all comments and suggestions. I suggest to editors accept it in the current form.